# Accuracy of frozen section in intraoperative margin assessment for breast-conserving surgery: A systematic review and meta-analysis

Mila Trementosa Garcia[1]*, Bruna Salani Mota[1], Natalia Cardoso[2], Ana Luiza Cabrera Martimbianco[3], Marcos Desidério Ricci[1], Filomena Marino Carvalho[4], Rodrigo Gonçalves[1], José Maria Soares Junior[1], José Roberto Filassi[1]

1 Department of Gynaecology/Obstetrics, University of São Paulo, São Paulo, Brazil, 2 Medical School, University of São Paulo, São Paulo, Brazil, 3 Graduate Program in Health and Environment, Universidade Metropolitana de Santos (UNIMES), Santos, Brazil, 4 Department of Pathology, University of São Paulo, São Paulo, Brazil

* Mila.trementosa@gmail.com

**Data Availability Statement:** All relevant data are within the manuscript and its Supporting Information files.

## Abstract

### Background and objectives

It is well established that tumor-free margin is an important factor for reducing local recurrence and reoperation rates. This systematic review with meta-analysis of frozen section intraoperative margin assessment aims to evaluate the accuracy, and reoperation and survival rates, and to establish its importance in breast-conserving surgery.

### Methods

A thorough review was conducted in all online publication-databases for the related literature up to March 2020. MeSH terms used: "Breast Cancer", "Segmental Mastectomy" and "Frozen Section". We included the studies that evaluated accuracy of frozen section, reoperation and survival rates. To ensure quality of the included articles, the QUADAS-2 tool (adapted) was employed. The assessment of publication bias by graphical and statistical methods was performed using the funnel plot and the Egger's test. The review protocol was registered in PROSPERO (CRD42019125682).

### Results

Nineteen studies were deemed suitable, with a total of 6,769 cases. The reoperation rate on average was 5.9%. Sensitivity was 0.81, with a Confidence Interval of 0.79–0.83, p = 0.0000, I2 = 95.1%, and specificity was 0.97, with a Confidence Interval of 0.97–0.98, p = 0.0000, I-2 = 90.8%, for 17 studies and 5,615 cases. Accuracy was 0.98. Twelve studies described local recurrence and the highest cumulative recurrence rate in 3 years was 7.5%. The quality of the included studies based on the QUADAS-2 tool showed a low risk of bias. There is no publication bias (p = 0.32) and the funnel plot showed symmetry.

**Funding:** The authors received no specific funding for this work.

**Competing interests:** No authors have competing interests.

## Conclusion

Frozen section is a reliable procedure with high accuracy, sensitivity and specificity in intraoperative margin assessment of breast-conserving surgery. Therefore, this modality of margin assessment could be useful in reducing reoperation rates.

## 1 Introduction

Breast-conserving surgery (BCS) followed by radiation therapy (RT) to eradicate microscopic residual disease is the standard procedure in early stage breast cancer treatment, since it provides similar survival rates, and better cosmetic results when compared to total mastectomy [1–4].

Reoperation rates in breast-conserving surgeries in literature range from 20% to 40% [5] due to positive margins status in H&E stain of the surgical specimen. The cause of such variation is multifactorial, but it is well-established that tumor-free margins excision reduces local recurrence and reoperation rates [6–11]. However, there is no consensus about the best method to achieve it, particularly intraoperative margin assessment. There are several techniques to evaluate intraoperative margins, such as gross analysis, radiography, cytology and frozen section procedure. Data from a cohort study, which included 24,217 patients, showed those that did not use frozen section during surgical procedures were four times more likely to need reoperation than women who underwent a lumpectomy for breast cancer followed by a frozen section procedure [12]. Despite the advantages of macroscopic analysis, this procedure can be performed directly by the surgeon, and boasts of higher accuracy (80%), sensitivity (49%) and specificity (86%) than other techniques [13].

The intraoperative frozen section analysis consists of selecting suspicious margins, freezing samples submitting them to histological sections, usually with the aid of a cryostat, and staining them for microscopy analysis. However, this implies an increase of surgery time [14–16], as well as the possibility of margin damage [17]. Furthermore, different studies did not reach a consensus regarding its accuracy and its impact on local recurrence rates.

Based on the abovementioned, we propose a systematic review with meta-analysis of intraoperative frozen section assessment of margins to analyze its accuracy when compared to final formalin-fixed paraffin embedded analysis, as well as reoperation and survival rates of patients submitted to this technique. The results of this review may help establishing the role of the frozen section assessment of margins in conserving surgeries.

## 2 Methods

This systematic review followed recommendations proposed by the Cochrane Handbook for Systematic Reviews of Diagnostic Test Accuracy [18] and the PRISMA statement (Preferred Reporting Items for Systematic Reviews and Meta-Analyses) [19]. The review protocol was registered and accepted by the international prospective register of systematic reviews (PROSPERO) under CRD42019125682.

### 2.1 Search methods for identification of studies

In March 2020, we conducted a systematic literature search of articles published on frozen section as a method employed for margin assessment on breast-conserving surgery using MEDLINE (via PubMed), Lilacs (via BVS), Embase (via Elsevier) and ClinicalTrials.gov, Cochrane

and "gray literature". No language and date restrictions were applied. MeSH terms: "Breast Cancer" [Title/Abstract], "Segmental Mastectomy" [Title/Abstract] and "Frozen Section" [Title/Abstract]. The search results were combined and exported to the EndNote® bibliographic management tool, and duplicate results were removed [19]. Two trained reviewers (M.T.G and N.C.) independently reviewed all titles for possible inclusion. All disagreements were resolved via consensus by a third senior researcher (B.S.M.).

## 2.2 Inclusion and exclusion criteria

All clinical trials and observational studies included this this review had the same type of target patients: women with invasive and/or *in situ* breast cancer that underwent breast-conserving surgery and had their margin samples submitted to frozen section assessment (index test). Only studies that presented certain data were included, such as outcome, accuracy compared to the formalin-fixed paraffin-embedded analysis (reference standard test), reoperation rates, and/or overall survival rate.

The exclusion criteria took into consideration overlapping databases, frozen section of only sentinel lymph nodes, no comparison with paraffin analysis or different methods of intraoperative assessment.

## 2.3 Data extraction

Two researchers manually extracted the following data from all studies included in this review: number of patients, number of cases, staging, age, concept of free margin, intraoperative margin assessment method, follow-up time, number of true positives (frozen section and paraffin with positive margins), number of true negatives (frozen section and paraffin with free margins), number of false positives (positive frozen section margins and free paraffin margins), number of false negatives (free frozen section margins and positive paraffin margins), total positive cases with the paraffin method, total negative cases with the paraffin method and re-excision rate. For local recurrence and overall survival, data was combined using the inverse variance method on the log-HR scale, and on the log-RR scale for dichotomous outcomes. If the data were diverse enough to permit effect sizes combination in a meaningful or valid manner, we presented such results individually using table and graphical formats, as well as a narrative approach to summarize the data. In cases where accuracy was not explicitly reflected, we constructed a 2 x 2 table to calculate the required data. All disagreements were resolved via consensus by a third senior reviewer.

## 2.4 Data collection and analysis

The next step was carried out by two reviewers, who screened all abstracts and potential articles to determine which would be submitted to a full manuscript evaluation. When a selected article lacked some necessary detail, including sensitivity and specificity, an attempt was made to contact the corresponding author.

## 2.5 Assessment of methodological quality

Two reviewers independently assessed quality of the articles using the QUADAS-2 tool [20] (University of Bristol, UK), adapted to this diagnostic accuracy meta-analysis. The resultant QUADAS-2 tool was used to assess studies in four key domains: patient selection, index test, reference standard, flow and timing. Questions in each domain were rated (low, high, unclear) in terms of risk of bias and concerns regarding applicability (for patient selection, index test

and reference standard only). All disagreements were resolved via consensus by a third expert researcher.

The assessment of publication bias by graphical and statistical methods was performed using the funnel plot and the Egger's test.

## 2.6 Statistical analysis and data synthesis

A meta-analysis was conducted using methods recommended by the Cochrane Handbook for Systematic Reviews of Diagnostic Test Accuracy. The accuracy of diagnostic tests was summarized by creating a 2 x 2 table for each study, based on information retrieved from the published papers. Test results were reported qualitatively (positive or negative) and their sensitivity and specificity (95% confidence intervals) were demonstrated in by forest plots created with the Review Manager 5 software to determine heterogeneity of diagnostic accuracy amongst included studies [21]. The receiver operating characteristic curve (SROC curves) was used to measure diagnostic performance. R version 3.1 and Meta-DiSc software were also employed to perform statistical analyses. A sensitivity and subgroup analysis were carried out, taking into consideration type of study, cut-off margin and histological subtype.

## 3 Results

In total, 2,298 studies were identified, manually cross-referenced and duplicate excluded. Of those, 2,262 were excluded since they did not fit the inclusion criteria, with 36 full-text evaluated articles remaining. Five were defined as "awaiting classification", while awaiting a reply to the contact emails sent to the corresponding authors, and twelve were excluded due to reasons described in Fig 1. In the end, 19 studies were deemed suitable for this review.

Study summaries are in Table 1. Thirteen papers were cohort studies [15–17, 22–31] and 6 were cross-sectional studies [32–37]. Those encompass a total of 6,769 frozen section assessments, including 3,811 from invasive ductal carcinoma (IDC) patients and 412 from ductal carcinoma *in situ* (DCIS) patients. Eighteen were conducted at tertiary center hospitals [15, 16, 22–32, 34–38], and one in a no-tertiary private healthcare center [17]. Definition of negative surgical margins ranged from "no ink on tumor" in 5 studies [17, 22, 26, 34, 35], 1 mm in 5 studies [16, 24, 30, 36, 37], 2 mm in 4 studies [25, 29, 31, 33] and 5 mm in 2 studies [15, 28]. Three studies evaluated tumor on cavity shaving margins [23, 27, 32]. Eleven out of 19 studies described the turnaround time necessary to perform the frozen section procedure; it ranged from 10 to 50 minutes.

For each study, patients that underwent frozen section were evaluated to collect accuracy measures such as true positive, true negative, false positive and false negative rates. Reoperation rates average was 5.9%, ranging from 0 to 23.9% (Table 2).

## 3.1 Sensitivity, specificity and SROC curves

Sensitivity and specificity were evaluated in 17 studies [15, 16, 22–29, 31–37]. Intraoperative assessment sensitivity was 0.81, with a CI of 0.79–0.83, p = 0.0000, and inconsistency ($I^2$) of 95.1%, which included the analysis of 5,615 tests in total (Fig 2). Specificity was 0.97, with a CI of 0.97–0.98, p = 0.0000, and inconsistency of 90.8% in the same sample (Fig 3). The accuracy, represented by the area under the SROC curve, is near to 1.0 (Fig 4).

**3.1.1 Sensitivity and subgroup analysis.** A sensitivity analysis was carried out considering only the cohort studies. Sensitivity and specificity were evaluated in 11 studies [15, 16, 22–29, 31]. Intraoperative assessment sensitivity was 0.87, with a CI of 0.85–0.89, p = 0.0000, and inconsistency ($I^2$) of 86.6%, which included a total of 4228 tests. Specificity was 0.97, with a CI

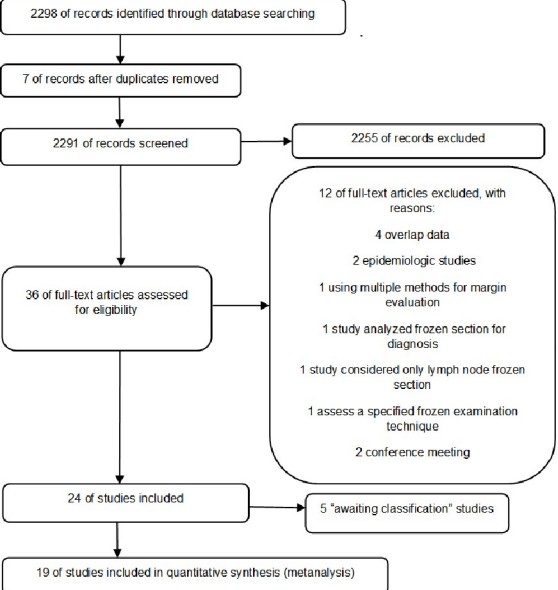

**Fig 1. PRISMA flowchart.**

of 0.96–0.97, p = 0.0000, and $I^2$ of 90.8% in the same sample. The accuracy, represented by the area under the SROC curve, is 0.98.

A sensitivity analysis was also carried out considering only the cross-section studies. Sensitivity and specificity were evaluated in 6 studies [32–37]. Intraoperative assessment sensitivity was 0.64, with a CI of 0.59–0.69), p = 0.0000, and inconsistency ($I^2$) of 97.1%, which included a total of 1387 tests. Specificity was 0.98, with a CI of 0.97–0.99, p = 0.0000, and $I^2$ of 91.5%, in the same sample. The accuracy, represented by the area under the SROC curve, is 0,98.

A sensitivity analysis was carried out considering only the 10 studies [15, 16, 24, 25, 28, 29, 31, 33, 36, 37] that evaluated margins ≥ 1mm. Sensitivity was 0.75, with a CI of 0.71–0.78), p = 0.0000, and $I^2$ = 96.8%, which included a total of 2,248 tests. Specificity was 0.96, with a CI of 0.95–0.97, p = 0.0000, and $I^2$ = 89.4%, in the same sample. SROC curve was 0.95.

The sensibility analysis by histological subtype was not possible due to lack of individual data on each test. Only two authors performed an evaluation by histological type, which will be describe in the results. Osako *et al.* showed an increase of 11.9 chance of positive margins in the final pathology (p = 0.01) in patients with invasive lobular carcinoma, larger tumors, or extensive intraductal component (EIC), and who were 50 years old or younger. Jorn *et al.* claimed that only disease multifocality (histologically discrete tumors at least 2 cm apart) could be a risk factor to increased reoperation rates, with OR of 3.41 (CI 1.38–8.40, p = 0.008). The article did not associate histological subtype and tumor sizer with further surgeries. The invasive ductal carcinoma subtype had an OR of 0.75 (CI 0.31–1.82, p = 0.37), invasive lobular carcinoma subtype had an OR of 2.29 (CI 0.52–9.98, p = 0.37) and larger tumor size (> 2 cm) OR 1.33 (CI 0.26–6.74, p = 0.733).

## 3.2 Local recurrence and survival

Twelve studies described local recurrence [15–17, 23–31]. It was not possible to perform a meta-analysis on these due to the lack of sufficient data to calculate hazard ratios.

In two studies, no patients presented local recurrence during an average follow-up of 40 months and 12 months, respectively [15, 16]. Caruso *et al.* (2011) had 1.9% of recurrence in

**Table 1. Summary description of included studies.**

| Author | Type of study | Country | Age (average) | Margin | Turnaround time | IDC | DCIS | IDC +DCIS | Mucinous | ILC | Mixt IDC +ILC | Other* |
|---|---|---|---|---|---|---|---|---|---|---|---|---|
| Anila 2016 | Cohort | India | 46 (23–71) | > 5 mm | 20 min | 50 | 2 | 7 | 1 | 0 | 0 | 0 |
| Caruso 2011 | Cohort | Italy | - | > 2 mm | 20 min | 33 | 2 | 6 | 0 | 7 | 0 | 4 |
| Cendán 2005 | Cross-sectional | USA | 59.4(48–60.8) | Tumor-bearing | 13 min | 57 | 33 | 0 | 0 | 7 | 0 | 0 |
| Dener 2009 | Cohort | Turkey | 49(18–94) | > 2 mm | 25 min | 170 | 0 | 0 | 0 | 16 | 4 | 0 |
| Ikeda 1997 | Cohort | Japan | 44.9 (33–66) | No ink on tumor | - | 47 | 9 | 0 | 0 | 0 | 0 | 0 |
| Jorns 2014 | Cross-sectional | USA | - | > 2 mm | 24 min | 23 | 20 | 0 | 0 | 2 | 0 | 1 |
| Kikuyama 2015 | Cohort | Japan | 51.2 (38–65) | No ink on tumor | - | 174 | 14 | 0 | 0 | 23 | 0 | 9 |
| Kim 2016 | Cohort | South Korea | 52.9 (-) | > 1 mm | - | 0 | 29 | 0 | 0 | 0 | 0 | 0 |
| Ko 2017 | Cross-sectional | South Korea | 50 (28–77) | No ink on tumor | 40 min | 420 | 63 | 0 | 0 | 14 | 0 | 12 |
| Noguchi 1995 | Cross-sectional | Japan | - | No ink on tumor | - | 85 | 13 | 0 | 1 | 0 | 0 | 1 |
| Nowikiewicz 2019 | Cross-sectional | Poland | 58.7 (25–85) | > 1mm | 15 min | 446 | 0 | 0 | 0 | 42 | 0 | 17 |
| Olson 2007 | Cohort | USA | 57.2 (27–89) | Tumor-bearing | 25 min | 214 | 33 | 0 | 7 | 17 | 1 | 20 |
| Osako 2015 | Cohort | Japan | | > 5 mm | 50 min | 794 | 142 | 0 | 0 | 33 | 0 | 46 |
| Pinotti 2002 | Cohort | Brazil | 53.7 (26–93) | > 2 mm | - | 81 | 0 | 0 | 4 | 8 | 1 | 6 |
| Riedl 2008 | Cohort | Austria | - | > 1 mm | 20–30 min | 901 | 0 | 0 | 0 | 115 | 0 | 0 |
| Rusby 2008 | Cohort | United Kingdom | 49.5 (40–58) | Tumor-bearing | 10–20 min | 81 | 6 | 0 | 0 | 11 | 0 | 17 |
| Sauter 1994 | Cross-sectional | USA | - | > 1 mm | - | 94 | 7 | 0 | 0 | 6 | 0 | 0 |
| Tan 2014 | Cohort | Singapore | 48 (28–78) | No ink on tumor | - | 108 | 18 | 0 | 0 | 5 | 0 | 7 |
| Weber 2008 | Cohort | Switzerland | 59.6 (33–86) | > 1 mm | - | 33 | 21 | 9 | 0 | 0 | 0 | 17 |
| TOTAL | | | | | | 3811 | 412 | 22 | 13 | 306 | 6 | 157 |

*Others: as authors describe or Paget´s disease; tubular, medullary, cribiform, papillary, apocrine, metaplastic, malignant fibrous histiocytoma.

IDC: Invasive ductal carcinoma / DCIS: Ductal carcinoma in situ / ILC: Invasive lobular carcinoma.

72.6 months of follow-up [31]. Dener et al. (2009) observed a 2.1% local recurrence rate with 62 months of follow-up [25]. Ikeda et al. (1997) found, in 3 years, a cumulative local recurrence rate of 7.5% [26]. Olson et al. (2007) had 2.7% of local recurrences during an average follow-up time of 53 months [27]. Osako et al. (2015), with an average follow-up time of 54.1 months, had 0.1% of breast cancer recurrence [28]. Pinotti et al. (2002) observed 1% of local recurrence at an average period of 42 months [29]. Tan et al. (2014) had 1.4% local recurrence in 45 months of follow-up [17]. Rusby et al. (2008) reported less than 1% in 41.4 months [23]. Weber et al. (2008) found a 5% local recurrence rate [24]. Lastly, Riedl et al. (2009) had an annual local recurrence rate of 1.2% [30].

Two studies compared overall survival between groups (re-excision and no re-excision) and no difference was found [25, 29]. Ikeda et al. (1997) had 100% overall survival after three years and 86% disease-free survival [26]. Osako et al. (2015) found, after 5 years, local recurrence free survival, disease-free survival and overall survival rates of 99.9%, 97.8%, and 98.2%, respectively [28]. And Caruso et al. (2011) had 98% of overall survival.

**Table 2. Frozen section results.**

| AUTHOR | PACIENTS | TESTS | TRUE POSITIVE | TRUE NEGATIVE | FALSE POSITIVE | FALSE NEGATIVE | REOPERACION/ PATIENTS (%) |
|---|---|---|---|---|---|---|---|
| ANILA 2016 | 60 | 60 | 40 | 20 | 0 | 0 | 0/60 (0%) |
| CARUSO 2011 | 50 | 53 | 5 | 44 | 3 | 1 | 0/50 (0%) |
| CENDÁN 2005 | 97 | 97 | 25 | 54 | 0 | 18 | 19/97 (19.5%) |
| DENER 2009 | 186 | 190 | 30 | 160 | 0 | 0 | 0/186 (0%) |
| IKEDA 1997 | 54 | 56 | 17 | 34 | 4 | 1 | 0/54 (0%) |
| JORNS 2014 | 46 | 46 | 12 | 28 | 0 | 6 | 11/46 (23.9%) |
| KIKUYAMA 2015* | 220 | 763 | 287 | 440 | 18 | 18 | - |
| KIM 2016 | 25 | 29 | 3 | 23 | 1 | 2 | 0/25 (0%) |
| KO 2017 | 509 | 483 | 120 | 338 | 1 | 24 | 32/509 (6.3%) |
| NOGUCHI 1995 | 95 | 100 | 23 | 64 | 12 | 1 | (0/95) |
| NOWIKIEWICZ 2019 | 505 | 505 | 4 | 429 | 0 | 72 | 72/505 (14.3%) |
| OLSON 2007* | 290 | 1311 | 57 | 1228 | 5 | 21 | 16/290 (5.5%) |
| OSAKO 2015 | 1029 | 1029 | 259 | 657 | 53 | 60 | 1/1029 (0.1%) |
| PINOTTI 2002 | 98 | 100 | 40 | 60 | 0 | 0 | - |
| RIEDL 2008 | 1016 | 1016 | - | - | - | 89 | 89/1016 (8,7%) |
| RUSBY 2008* | 115 | 557 | 39 | 495 | 15 | 8 | 3/115 (2.6%) |
| SAUTER 1994 | 107 | 156 | 40 | 107 | 4 | 5 | - |
| TAN 2014 | 138 | 138 | - | - | - | 0 | 0/138 (0%) |
| WEBER 2008 | 78 | 80 | 32 | 35 | 5 | 8 | 10/78 (12%) |
| TOTAL | 4718 | 6769 | | | | | 253/4293 (5.9%) |

*Analysis for each specimen margin.

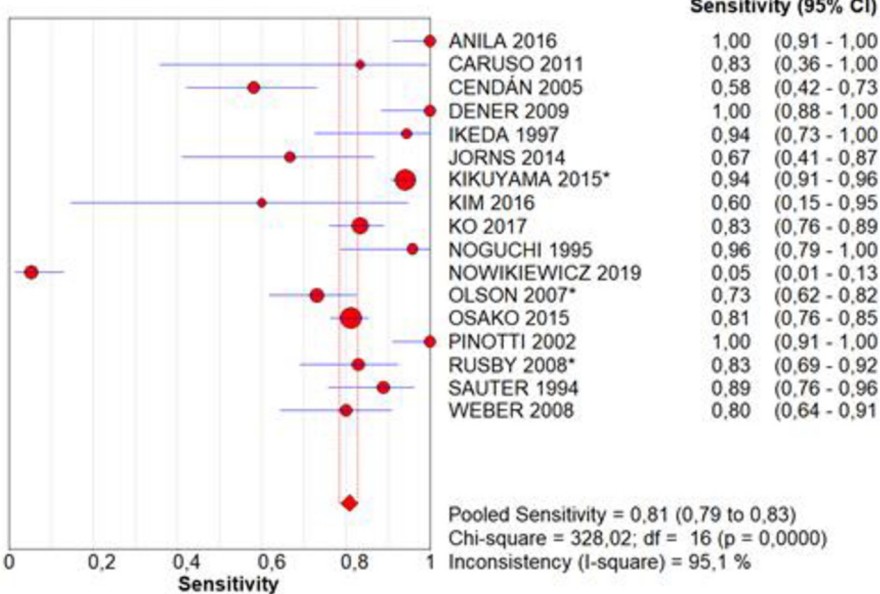

**Fig 2. Sensitivity.**

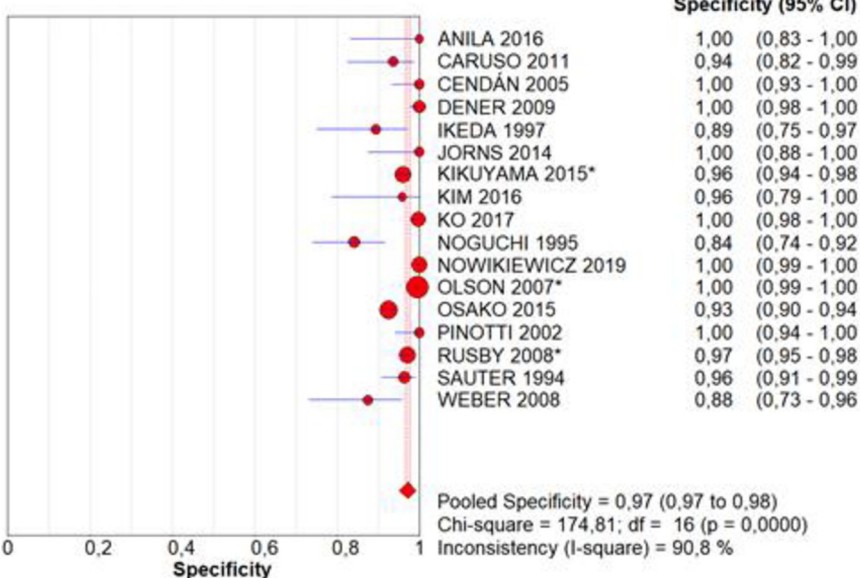

**Fig 3. Specificity.**

## 3.3 Methodological quality of included studies

Using the adapted QUADAS-2 tool, the risk of bias was analyzed in each selected study (Fig 5).

Regarding participant selection, studies were considered to present low risk of bias since all studies included only patients with previous breast cancer diagnosis.

Regarding index test, an unclear risk of bias was determined for 18 included studies due to intrinsic subjectivity of pathologists [15–17, 22–36]. Only Sauter *et al.* (1994) compared

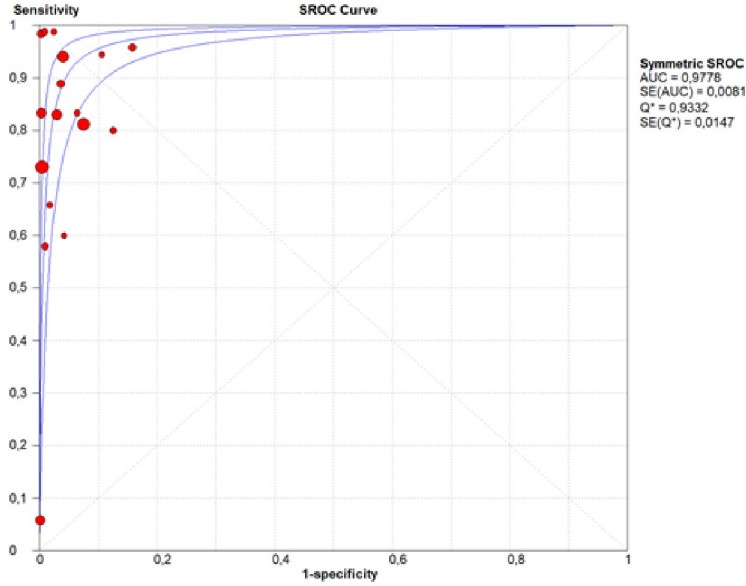

**Fig 4. SROC curve.**

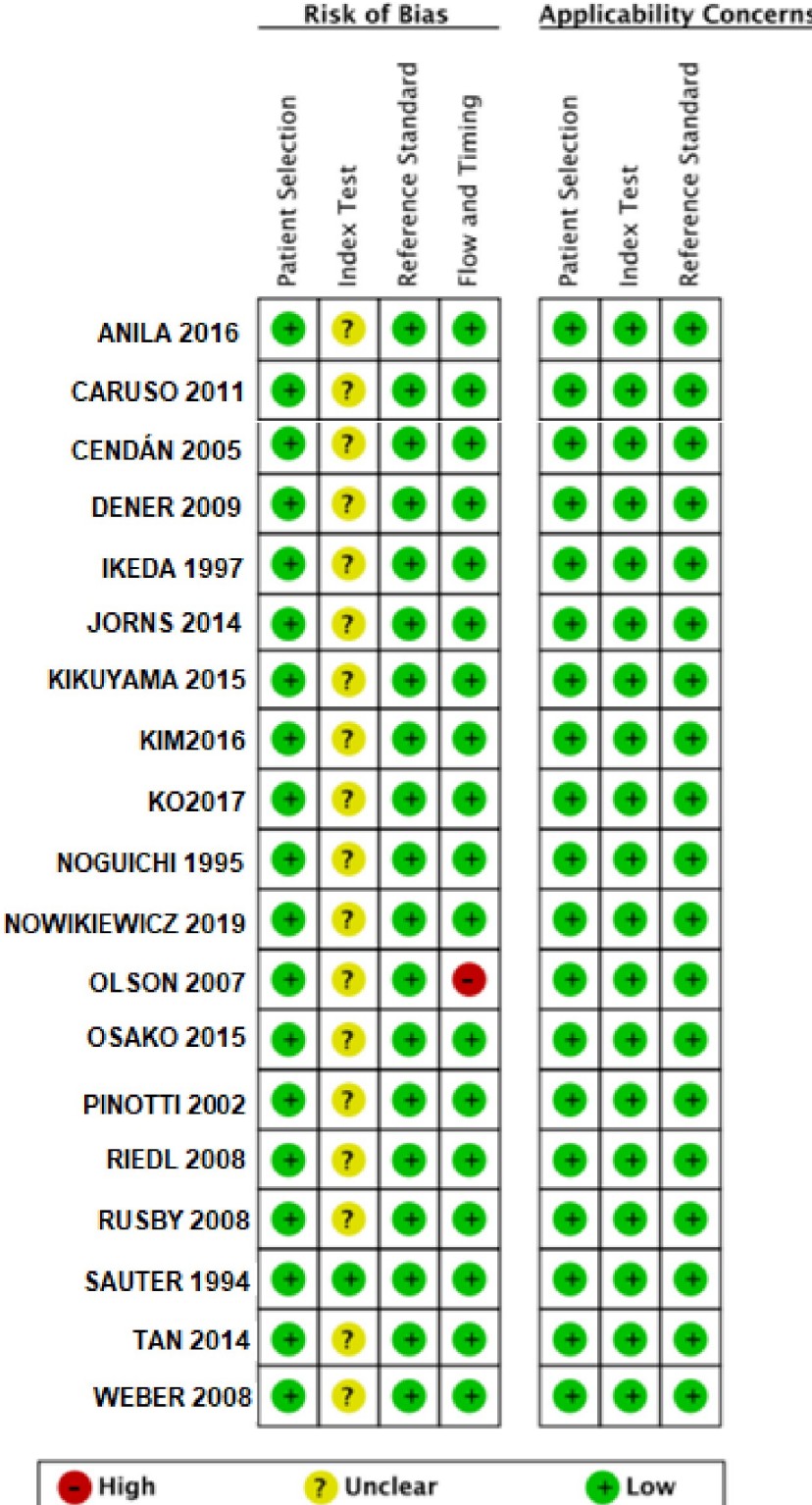

**Fig 5. Risk of bias by QUADAS-2.**

accuracy of each pathologist in frozen section assessment, therefore this study was considered as a low risk of bias for this modality [37].

In the flow and timing assessment, 18 out 19 studies were considered as having low risk of bias [15–17, 22–26, 28–37]. Olson *et al.* (2007) was considered as high risk of bias due to inadequate exclusion [27].

There was no publication bias (p 0.32) and the funnel plot showed symmetry (Fig 6).

## 4 Discussion

Despite the large variability of negative margin definitions, it is well-known that positive margins in breast-conserving surgeries are associated with increased rates of local recurrence [39].

Reducing reoperation rates is the greatest advantage of intraoperative frozen section margin assessment, which consequently reduces patient anxiety and improves quality of life. Moreover, with the increase in BCS, more favorable cosmetic outcomes are made possible, sometimes preventing mastectomy altogether. This saves money on additional surgeries and hospital stays, and avoids delays in the start of adjuvant treatments. Main limitations relate to technical difficulties of the method, availability of a pathologist in the operating room, increased costs and additional time in the operating room.

Some oncology centers routinely perform the intraoperative assessment of the margins with frozen section and/or touch cytological imprint (TIC). A meta-analysis study, which includes 9 studies related to frozen section, evaluated the accuracy of different intraoperative techniques for margin assessment and reported sensitivity of 86% and specificity of 91% with 97% of heterogeneity for the frozen section technique [40]. Our sample, which it is 50% larger (n = 4,293 exams), has shown a slightly lower sensitivity (81%), with higher specificity 97% for the frozen section method, but still with a high risk of inconsistency ($I^2$ = 90.8%). This might be due to the setting in which the included studies were carried out, all in tertiary centers, which probably implies the pathologists and surgeons are more experienced.

Our meta-analysis is novel in the sense that a methodological quality assessment of studies was included using the QUADAS-2 tool, thus associating the frozen section test to breast-conserving surgery and reoperation rates. Another strength of this study is the use of the Cochrane Handbook for Systematic Reviews of Diagnostic Test Accuracy.

This study has some limitations, though, which are intrinsic to the quality of the included studies due to heterogeneity of the available data, including the definition of free margins, no reply from e-mails requesting raw data, and lack of stratified data of true positives, false positives, true negative and false negatives for DCIS and IDC.

For patients with invasive tumors, a consensus statement (2014) has suggested that a positive margin should be considered as "tumor at ink" [41]. Less than 1 mm of histologically normal tissue between the tumor and the resected border can be considered "clear" and therefore, do not require re-excision. This consensus also considered this margin equally appropriate for patients with *in situ* tumors, and associated with invasive carcinoma, as long as the intraductal component is smaller than 25 percent of the tumor. Since 2013, a trend in the reduction of reoperation rates has been observed, which was described by Yang *et al.* [42]. Therefore, in 2016, Morrow *et al.* showed a decrease of 16% in re-excision rates among surgeons consensus [43]. For patients with exclusive ductal carcinoma *in situ* (DCIS), the National Comprehensive Cancer Network (NCCN) guidelines had previously suggested a margin of ≥ 1 mm for DCIS, which could increase re-excision rates if compared to the definition of negative margin as "no tumor at ink" [44]. In this review, it was not possible to perform separate analysis of IDC and DCIS. Even if studies included both neoplasias, none presented separate accuracies for each. Cabioglu *et al.* (2007) reported reoperation rates among DCIS twice as high (14%) when

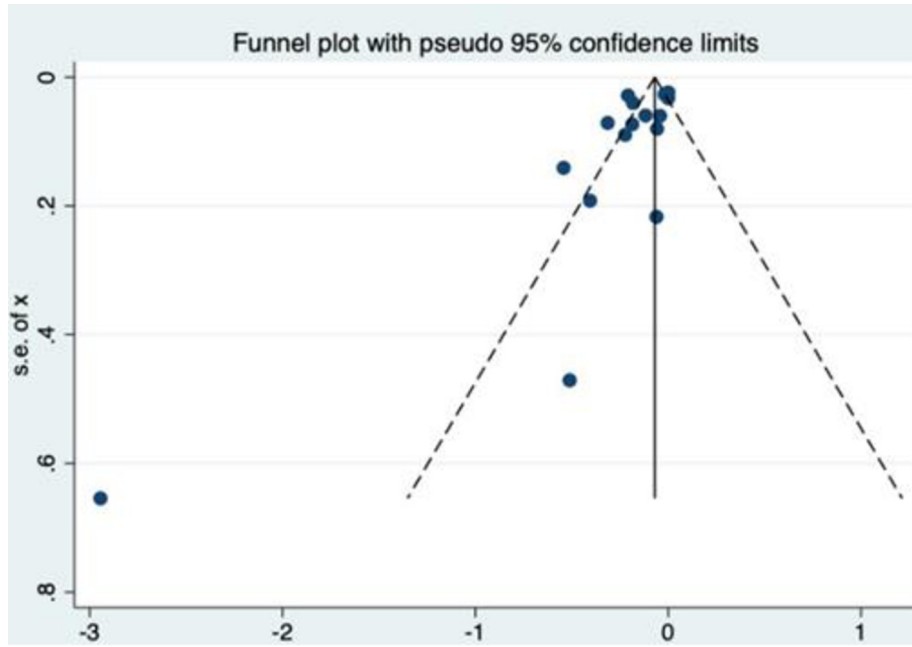

**Fig 6. Funnel plot of publication bias.**

compared to IDC (7%) [45]. This is the core issue of this review and may influence future guidelines since it could possibly be incorporated into clinical practices.

Analyzing some older studies, we are left with different definitions of free margins [39]. When a larger margin is required to be considered free, this could interfere in true positive, true negative, false positive and false negative rates and, therefore, would also interfere with the accuracy of the technique.

Five studies were left as "awaiting classification", since attempts to contact corresponding authors by email to obtain their stratified data regarding accuracy received no reply, and thus means data could not be extracted.

In clinical practice, avoiding readmission and reoperation would decrease hospital expenses; in that sense, Alvarado *et al* estimated that frozen section assessments could result in an yearly saving of $3.7 billion, which means less than $20,000/QALY (quality-adjusted life years) and a 89.7% reduction of reoperation rates.

Despite false negative rates of up to 23%, the reoperation rate found is still much lower than expected and this might be due to the great variability in the interpretation of test results among the studies. Ikeda *et al.* (1997) opted for radiotherapy for false negative cases based on patient's opinion and physician's advice [26]. Kim *et al.* defined positive margins as > 1mm, however they did not reoperate false positive cases because cancer cells were not in the margin itself [16]. Only one patient with a false negative result in the Noguchi study refused a second operation because since the involvement was histologically minimal [35]. Osako *et al.* (2015) did not reoperate 59 out of 60 false negative cases due to minimal residual disease [28].

This review, considering only studies that analyzed LR, found rates ranging from 0 to 7.5% in a follow-up average of 12–62 months. Local recurrence rate (LR) of 4.2% was reported for overall breast-conserving surgeries [46].

In the future, the findings of this meta-analysis will be used as the parameters required for the development of a Markov model to determine whether the implementation of intraoperative frozen section assessments in the Brazilian public health system is a cost-effective

intervention. Since studies from different countries were included, this model could easily be adapted to other settings, private or public, in different countries, improving health care services at adequate costs.

## 5 Conclusion

Frozen section is a reliable technique for intraoperative margin assessment in breast-conserving surgery with high levels of accuracy, sensitivity and specificity. Due to this high precision for negative results, routine use of this test may aid surgeons in the pursuit of tumor-free surgical margins, therefore reducing reoperation rates.

## Supporting information

**S1 Checklist. PRISMA 2009 checklist.**
(DOC)

**S1 Table.**
(XLSX)

## Acknowledgments

I would like to thank my department colleagues for all the support.

## Author Contributions

**Conceptualization:** Mila Trementosa Garcia.

**Data curation:** Natalia Cardoso.

**Formal analysis:** Ana Luiza Cabrera Martimbianco.

**Methodology:** Bruna Salani Mota.

**Project administration:** Marcos Desidério Ricci.

**Software:** Rodrigo Gonçalves.

**Supervision:** Filomena Marino Carvalho.

**Visualization:** José Maria Soares Junior.

**Writing – original draft:** Mila Trementosa Garcia.

**Writing – review & editing:** José Roberto Filassi.

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
