## [Decision Letter · Decision Letter 0]

21 Dec 2020

PONE-D-20-35344

Accuracy of frozen section in intra-operative margin assessment in breast conserving surgery: systematic review and metanalysis

PLOS ONE

Dear Dr. garcia,

Thank you for submitting your manuscript to PLOS ONE. After careful consideration, we feel that it has merit but does not fully meet PLOS ONE’s publication criteria as it currently stands. Therefore, we invite you to submit a revised version of the manuscript that addresses the points raised during the review process.

We look forward to receiving your revised manuscript.

Kind regards,

Lanjing Zhang, MD, MS

Academic Editor

PLOS ONE

Journal Requirements:

3. Please confirm that you have included all items recommended in the PRISMA checklist including the full electronic search strategy used to identify studies with all search terms and limits for at least one database.

4. Please described the data extraction methods in more details. We would expect to see reporting of the specific information extracted from the manuscripts.

5. Please include your tables as part of your main manuscript and remove the individual files. Please note that supplementary tables should be uploaded as separate "supporting information" files.

6. Please include captions for your Supporting Information files at the end of your manuscript, and update any in-text citations to match accordingly. Please see our Supporting Information guidelines for more information: http://journals.plos.org/plosone/s/supporting-information

Additional Editor Comments:

This is a well-designed systematic review and meta-analysis. The findings are not very novel as one of the reviewers commented, while its method is overall acceptable. See the following concerns for your consideration.

Major points:

1. The search terms "lumpectomy" and "breast conserving surgery" are missing, but should be included.

2. Does frozen section also have high sensitivity? If so, please comment on it and add "sensitivity" in the concluding sentence.

3. Cross-sectional and cohort studies have different study design and scientific rigor/biases. It will be very important and interesting to conduct subgroup analyses based on study design (Cross-sectional and cohort studies in different groups).

4. Histology types of breast cancer may influence its outcome, and cancer distribution pattern. So it may also be necessary either discuss this point in light of/citing prior studies ((e.g. PMID: 28746732 and PMID: 30518616) or conduct subgroup analysis.

Minor points:

1. It will be helpful if the sensitivity and specificity of each study are shown.

2. Recommend to cite the papers that shows increasing use of lumpectomy/BCS in recent years (e.g. PMID: 24929768, PMID: 29423512 and PMID: 28586788)

3. Writing style issues: Just some examples.

Title: in breast ... may be replaced with for breast...

Abstract: metanalysis and exams may be replaced with meta-analysis and studies, respectively; CI should be spelled out, I2 may be replaced with I-squared or I-2;

Section 2.2: "we admitted" may be replaced with "we included"

Section 2.4: a "that" is needed before "were selected"

Reviewers' comments:

Reviewer's Responses to Questions

**Comments to the Author**

1. Is the manuscript technically sound, and do the data support the conclusions?

Reviewer #1: Partly

Reviewer #2: Partly

2. Has the statistical analysis been performed appropriately and rigorously? 

Reviewer #1: I Don't Know

Reviewer #2: N/A

3. Have the authors made all data underlying the findings in their manuscript fully available?

Reviewer #1: Yes

Reviewer #2: Yes

4. Is the manuscript presented in an intelligible fashion and written in standard English?

Reviewer #1: No

Reviewer #2: Yes

5. Review Comments to the Author

Reviewer #1: This manuscript demonstrated results of systematic review and metanalysis on accuracy of frozen section on margin evaluation in breast conserving surgery.

1. This topic is not cutting edge; it was extensively discussed on the literature more than 10 years ago on this breast pathology practice. Currently,

The frozen section for margin evaluation in partial mastectomy specimens is not commonly used in an intraoperative setting of pathology practice at least in hospitals located in the US due to some technical issues including but not limited to the following:

-It is well known that breast tissue is fatty and difficult to cut into a complete section for microscopic evaluation when

it is fresh (without fixed in formalin solution for certain period of time). This nature of fresh breast tissue makes

margin evaluation by frozen section often challenging. Nowadays, some surgeons use radiographic image on fresh

tissue specimen instead of frozen sections to decide if margins are clear and if a new margin (or which margin) needs

to be re-excised before completing a breast conserving surgery. This practice has been gone well over the years in

our institution.

-Also sometimes cutting frozen section on fresh breast tissue takes a longer time due to fatty nature of the specimen

compared to cutting non-fatty tissues; furthermore the frozen section practice prolongs operating time.

Although this study showed an accuracy of frozen section in intra-operative margin assessment in breast conserving surgery, now this practice has not been widely adopted at least in the US, probably, partly due to the reasons mentioned above. Therefore, not sure if this study had significant clinical utility at the current breast conserving surgery practice across the world.

2. A total of 19 studies (42 references) were analyzed; not sure if number of the studies is enough for this metanalysis.

3. English needs to be polished; there are some grammatical errors.

Reviewer #2: 1. Some hospitals routinely perform intraoperative margin assessment in breast conserving surgery, while others do not. Can you compare advantage and disadvantage of different techniques, including gross analysis, radiology, and cytology examination? How is local recurrence and reoperation rate individually? What scenarios do authors prefer to do frozen sections rather than other methods?

2. Is there any difference about false positive rate and false negative rate among different cancer type, such as IDC, ILC or mucinous carcinoma?

3. There is the large variability in false negative rate from 0% to 23%. What are reasons to cause this?

4. In Jorns and Ko studies, the number of patients with reoperation is larger than the number of patients with false negative rates. Except false negative margin, is there any other reason to let surgeons re-operate patients?

5. Is it possible to list turnaround time for results in the table considering that one of disadvantages of intraoperative assessment is to increase surgery time?

6. PLOS authors have the option to publish the peer review history of their article (what does this mean?). If published, this will include your full peer review and any attached files.

Reviewer #1: No

Reviewer #2: No

---

## [Author Response · Author response to Decision Letter 0]

2 Feb 2021

REVIEWER 1:

- Thank you for your consideration. We know that frozen section is not a common procedure for breast cancer in United States, but it is still use worldwide. Data from a cohort study, including 24.217 patients, showed four times more chance of reoperation without frozen section during surgical procedures (American College of Surgeons National Surgical Quality Improvement Program – NSQIP) than women who undergo a lumpectomy for breast cancer surgery with frozen section exam (Mayo Clinic Rochester). We added it in Introduction. 

- A total of 19 studies (42 references) were analyzed; not sure if number of the studies is enough for this metanalysis.

We perform a rigorous search strategy to find the studies related to the frozen section in breast cancer surgery. We are confident that this search strategy covers all studies that could have available data to perform it. Here we have 19 studies including 6760 participants.

- English needs to be polished; there are some grammatical errors.

Thank you. We performed a grammar review with a specialist. 

REVIEWER 2:

- Some hospitals routinely perform intraoperative margin assessment in breast conserving surgery, while others do not. Can you compare advantage and disadvantage of different techniques, including gross analysis, radiology, and cytology examination? How is local recurrence and reoperation rate individually? What scenarios do authors prefer to do frozen sections rather than other methods?

 We added the paragraph in Introduction:

Despite the advantages of macroscopic analysis, this procedure can be performed directly by the surgeon, and boasts of higher accuracy (80%), sensitivity (49%) and specificity (86%) than other techniques.

- Is there any difference about false positive rate and false negative rate among different cancer type, such as IDC, ILC or mucinous carcinoma?

We added a paragraph at 3.1.1 Sensitivity and subgroup analysis:

The sensibility analysis by histological subtype was not possible due to lack of individual data on each test. Only two authors performed an evaluation by histological type, which will be describe in the results. Osako et al. showed an increase of 11.9 chance of positive margins in the final pathology (p=0.01) in patients with invasive lobular carcinoma, larger tumors, or extensive intraductal component (EIC), and who were 50 years old or younger. Jorn et al. claimed that only disease multifocality (histologically discrete tumors at least 2 cm apart) could be a risk factor to increased reoperation rates, with OR of 3.41 (CI 1.38-8.40, p=0.008). The article did not associate histological subtype and tumor sizer with further surgeries. The invasive ductal carcinoma subtype had an OR of 0.75 (CI 0.31-1.82, p=0.37), invasive lobular carcinoma subtype had an OR of 2.29 (CI 0.52-9.98, p=0.37) and larger tumor size (> 2 cm) OR 1.33 (CI 0.26-6.74, p=0.733).

- There is the large variability in false negative rate from 0% to 23%. What are reasons to cause this?

The reason for the large variability could be associated with the pathologist's experience. It is not possible to identify in the studies the physician's expertise. Moreover, a different definition of free margin between studies could explain this large variability.

- In Jorns and Ko studies, the number of patients with reoperation is larger than the number of patients with false negative rates. Except false negative margin, is there any other reason to let surgeons re-operate patients?

Jorns study: 11 patients were reoperated. Among those 5 were true positive and 6 were false negative at frozen section which led the patient to another surgery. Based on this all patients were submitted to another surgery based only in the frozen section exam. 

 Ko study: considered some frozen sections as undetermined margins. This kind of margins were not evaluated even as positive neither negative. This factor increases the number of reoperation rate.

- Is it possible to list turnaround time for results in the table considering that one of disadvantages of intraoperative assessment is to increase surgery time?

Eleven out of 19 studies described turnaround time to perform the frozen section exam, it ranges from 10 to 50 minutes. We added a column in table 1.

---

## [Decision Letter · Decision Letter 1]

16 Feb 2021

PONE-D-20-35344R1

Accuracy of frozen section in intraoperative margin assessment for breast-conserving surgery: a systematic review and meta-analysis

PLOS ONE

Dear Dr. garcia,

Thank you for submitting your manuscript to PLOS ONE. After careful consideration, we feel that it has merit but does not fully meet PLOS ONE’s publication criteria as it currently stands. Therefore, we invite you to submit a revised version of the manuscript that addresses the points raised during the review process.

Please specifically address the editor's comments which seemed not be done in the first round of revision. 

We look forward to receiving your revised manuscript.

Kind regards,

Lanjing Zhang, MD, MS

Academic Editor

PLOS ONE

Additional Editor Comments (if provided):

The revision has greatly improved the manuscript, but my concerns seemed not be addressed at all (See below). The abstract and text still have some writing style issues.

This is a well-designed systematic review and meta-analysis. The findings are not very novel as one of the reviewers commented, while its method is overall acceptable. See the following concerns for your consideration.

Major points:

1. The search terms "lumpectomy" and "breast conserving surgery" are missing, but should be included.

2. Does frozen section also have high sensitivity? If so, please comment on it and add "sensitivity" in the concluding sentence.

3. Cross-sectional and cohort studies have different study design and scientific rigor/biases. It will be very important and interesting to conduct subgroup analyses based on study design (Cross-sectional and cohort studies in different groups).

4. Histology types of breast cancer may influence its outcome, and cancer distribution pattern. So it may also be necessary either discuss this point in light of/citing prior studies ((e.g. PMID: 28746732 and PMID: 30518616) or conduct subgroup analysis.

Minor points:

1. It will be helpful if the sensitivity and specificity of each study are shown.

2. Recommend to cite the papers that shows increasing use of lumpectomy/BCS in recent years (e.g. PMID: 24929768, PMID: 29423512 and PMID: 28586788)

3. Writing style issues: Just some examples.

Title: in breast ... may be replaced with for breast...

Abstract: metanalysis and exams may be replaced with meta-analysis and studies, respectively; CI should be spelled out, I2 may be replaced with I-squared or I-2;

Section 2.2: "we admitted" may be replaced with "we included"

Section 2.4: a "that" is needed before "were selected"

I would also modify the abstract as:

Background and objectives: It is well established that tumor-free margins is an important factor

for reducing local recurrence and reoperation rates. This systematic review with meta-analysis of

frozen section intraoperative margin assessment aims to evaluate the accuracy, and reoperation

and survival rates, and to establish its importance in breast-conserving surgery. Methods: A thorough review

was conducted in all online publication-databases for the related literature up to March 2020. MeSH terms used: “Breast

Cancer”, “Segmental Mastectomy” and “Frozen Section”. We included the studies that evaluated accuracy of

frozen section, reoperation and survival rates. To ensure quality of the included articles, the

QUADAS-2 tool (adapted) was employed. The assessment of publication bias by graphical and

statistical methods was performed using the funnel plot and the Egger’s test. The review protocol

was registered in PROSPERO (CRD42019125682). Results: Nineteen studies were deemed

suitable, with a total of 6,769 cases. The reoperation rate on average was 5.9%. Sensitivity was 0.81,

with a Confidence Interval of 0.79–0.83, p=0.0000, I-2=95.1%, and specificity was 0.97, with a

Confidence Interval of 0.97–0.98, p=0.0000, I-2 =90.8%, for 17 studies and 5,615 cases. Accuracy

was 0.98. Twelve studies described local recurrence and the highest cumulative recurrence rate

in 3 years was 7.5%. The quality of the included studies based on the QUADAS-2 tool showed a

low risk of bias. There is no publication bias (p=0.32) and the funnel plot showed symmetry.

Conclusion: Frozen section is a reliable procedure with high accuracy, sensitivity and specificity

in intraoperative margin assessment of breast-conserving surgery. Therefore, this modality of

margin assessment could be useful in reducing reoperation rates.

Reviewers' comments:

Reviewer's Responses to Questions

**Comments to the Author**

1. If the authors have adequately addressed your comments raised in a previous round of review and you feel that this manuscript is now acceptable for publication, you may indicate that here to bypass the “Comments to the Author” section, enter your conflict of interest statement in the “Confidential to Editor” section, and submit your "Accept" recommendation.

Reviewer #2: All comments have been addressed

2. Is the manuscript technically sound, and do the data support the conclusions?

Reviewer #2: Yes

3. Has the statistical analysis been performed appropriately and rigorously? 

Reviewer #2: N/A

4. Have the authors made all data underlying the findings in their manuscript fully available?

Reviewer #2: Yes

5. Is the manuscript presented in an intelligible fashion and written in standard English?

Reviewer #2: Yes

6. Review Comments to the Author

Reviewer #2: (No Response)

7. PLOS authors have the option to publish the peer review history of their article (what does this mean?). If published, this will include your full peer review and any attached files.

Reviewer #2: No

---

## [Author Response · Author response to Decision Letter 1]

2 Mar 2021

# EDITOR:

Major points:

1. The search terms "lumpectomy" and "breast conserving surgery" are missing but should be included.

The search terms "lumpectomy" and "breast conserving surgery" are included at MESH tree in PubMed for segmental mastectomy.

2. Does frozen section also have high sensitivity? If so, please comment on it and add "sensitivity" in the concluding sentence.

After the sensitivity analysis considering only the cohort studies, the test's sensitivity increased from 80 to 90%. We ended up at the conclusion:

Frozen section is a reliable technique for intraoperative margin assessment in breast-conserving surgery with high levels of accuracy, sensitivity and specificity. Due to this high precision for negative results, routine use of this test may aid surgeons in the pursuit of tumor-free surgical margins, therefore reducing reoperation rates.

3. Cross-sectional and cohort studies have different study design and scientific rigor/biases. It will be very important and interesting to conduct subgroup analyses based on study design (Cross-sectional and cohort studies in different groups).

We performed analyses considering only cohort and cross-sectional studies. There was an increase in sensitivity. We described it in the text and added the figures (3.1.1 Sensitivity and subgroup analysis):

A sensitivity analysis was carried out considering only the cohort studies. Sensitivity and specificity were evaluated in 11 studies(15,16,22–29,31). Intraoperative assessment sensitivity was 0.87, with a CI of 0.85–0.89, p=0.0000, and inconsistency (I2) of 86.6%, which included a total of 4228 tests (Figure 5). Specificity was 0.97, with a CI of 0.96–0.97, p=0.0000, and I2 of 90.8% in the same sample (Figure 6). The accuracy, represented by the area under the SROC curve, is 0.98 (Figure 7).

A sensitivity analysiss was also carried out considering only the cross-section studies. Sensitivity and specificity were evaluated in 6 studies(32–37). Intraoperative assessment sensitivity was 0.64, with a CI of 0.59–0.69), p=0.0000, and inconsistency (I2) of 97.1%, which included a total of 1387tests (Figure 8). Specificity was 0.98, with a CI of 0.97–0.99, p=0.0000, and I2 of 91.5%, in the same sample (Figure 9). The accuracy, represented by the area under the SROC curve, is 0,98 (Figure 10).

4. Histology types of breast cancer may influence its outcome, and cancer distribution pattern. So it may also be necessary either discuss this point in light of/citing prior studies ((e.g. PMID: 28746732 and PMID: 30518616) or conduct subgroup analysis.

Unfortunately, the studies did not allow the extraction of individual data about accuracy by histological type. Only 2 authors performed an evaluation by histological type. We described it in the results. The authors Yang 2017 and Metzger-Filho 2019 described the overall survival versus histological data. 

We added a paragraph at 3.1.1 Sensitivity and subgroup analysis:

The sensibility analysis by histological subtype was not possible due to lack of individual data on each test. Only two authors performed an evaluation by histological type, which will be describe in the results. Osako et al. showed an increase of 11.9 chance of positive margins in the final pathology (p=0.01) in patients with invasive lobular carcinoma, larger tumors, or extensive intraductal component (EIC), and who were 50 years old or younger. Jorn et al. claimed that only disease multifocality (histologically discrete tumors at least 2 cm apart) could be a risk factor to increased reoperation rates, with OR of 3.41 (CI 1.38-8.40, p=0.008). The article did not associate histological subtype and tumor sizer with further surgeries. The invasive ductal carcinoma subtype had an OR of 0.75 (CI 0.31-1.82, p=0.37), invasive lobular carcinoma subtype had an OR of 2.29 (CI 0.52-9.98, p=0.37) and larger tumor size (> 2 cm) OR 1.33 (CI 0.26-6.74, p=0.733).

Minor points:

1. It will be helpful if the sensitivity and specificity of each study are shown.

These were shown at figure 2 e 3.

2. Recommend to cite the papers that shows increasing use of lumpectomy/BCS in recent years (e.g. PMID: 24929768, PMID: 29423512 and PMID: 28586788)

 The PMID: 24929768 was added in introduction: 

Data from a cohort study, which included 24,217 patients, showed those that did not use frozen section during surgical procedures were four times more likely to need reoperation than women who underwent a lumpectomy for breast cancer followed by a frozen section procedure. Despite the advantages of macroscopic analysis, this procedure can be performed directly by the surgeon, and boasts of higher accuracy (80%), sensitivity (49%) and specificity (86%) than other techniques 

The PMID: 28586788 and PMID: 29423512 was added in discussion:

Since 2013, a trend in the reduction of reoperation rates has been observed, which was described by Yang et al. Therefore, in 2016, Morrow et al. showed a decrease of 16% in re-excision rates among surgeons consensus.

3. Writing style issues: Just some examples.

Title: in breast ... may be replaced with for breast...

Ok. Thank you.

Abstract: metanalysis and exams may be replaced with meta-analysis and studies, respectively; CI should be spelled out, I2 may be replaced with I-squared or I-2;

Ok. Thank you.

We preferred replaced exam to test, to prevent studies = papers due to our unit analysis was performed by test and not only by number the included patient.

Section 2.2: "we admitted" may be replaced with "we included"

Ok. Thank you.

Section 2.4: a "that" is needed before "were selected"

Ok. Thank you.

I would also modify the abstract as: 

Background and objectives: It is well established that tumor-free margins is an important factor for reducing local recurrence and reoperation rates. This systematic review with meta-analysis of frozen section intraoperative margin assessment aims to evaluate the accuracy, and reoperation and survival rates, and to establish its importance in breast-conserving surgery. Methods: A thorough review was conducted in all online publication-databases for the related literature up to March 2020. MeSH terms used: “Breast Cancer”, “Segmental Mastectomy” and “Frozen Section”. We included the studies that evaluated accuracy of frozen section, reoperation and survival rates. To ensure quality of the included articles, the QUADAS-2 tool (adapted) was employed. The assessment of publication bias by graphical and statistical methods was performed using the funnel plot and the Egger’s test. The review protocol was registered in PROSPERO (CRD42019125682). Results: Nineteen studies were deemed suitable, with a total of 6,769 cases. The reoperation rate on average was 5.9%. Sensitivity was 0.81, with a Confidence Interval of 0.79–0.83, p=0.0000, I2=95.1%, and specificity was 0.97, with a Confidence Interval of 0.97–0.98, p=0.0000, I-2 =90.8%, for 17 studies and 5,615 cases. Accuracy was 0.98. Twelve studies described local recurrence and the highest cumulative recurrence rate in 3 years was 7.5%. The quality of the included studies based on the QUADAS-2 tool showed a low risk of bias. There is no publication bias (p=0.32) and the funnel plot showed symmetry. Conclusion: Frozen section is a reliable procedure with high accuracy, sensitivity and specificity in intraoperative margin assessment of breast-conserving surgery. Therefore, this modality of margin assessment could be useful in reducing reoperation rates.

Keywords: breast cancer; conserving surgery; frozen section; diagnostic accuracy; margin assessment; intraoperative test

# REVIEWER 1

This manuscript demonstrated results of systematic review and metanalysis on accuracy of frozen section on margin evaluation in breast conserving surgery.

1. This topic is not cutting edge; it was extensively discussed on the literature more than 10 years ago on this breast pathology practice. Currently, the frozen section for margin evaluation in partial mastectomy specimens is not commonly used in an intraoperative setting of pathology practice at least in hospitals located in the US due to some technical issues including but not limited to the following:

-It is well known that breast tissue is fatty and difficult to cut into a complete section for microscopic evaluation when it is fresh (without fixed in formalin solution for certain period of time). This nature of fresh breast tissue makes margin evaluation by frozen section often challenging. Nowadays, some surgeons use radiographic image on fresh tissue specimen instead of frozen sections to decide if margins are clear and if a new margin (or which margin) needs

to be re-excised before completing a breast conserving surgery. This practice has been gone well over the years in our institution. 

-Also sometimes cutting frozen section on fresh breast tissue takes a longer time due to fatty nature of the specimen compared to cutting non-fatty tissues; furthermore the frozen section practice prolongs operating time. Although this study showed an accuracy of frozen section in intra-operative margin assessment in breast conserving surgery, now this practice has not been widely adopted at least in the US, probably, partly due to the reasons mentioned above. Therefore, not sure if this study had significant clinical utility at the current breast conserving surgery practice across the world.

 Thank you for your consideration. We know that frozen section is not a common procedure for breast cancer in United States, but it is still use worldwide. Data from a cohort study, including 24.217 patients, showed four times more chance of reoperation without frozen section during surgical procedures (American College of Surgeons National Surgical Quality Improvement Program – NSQIP) than women who undergo a lumpectomy for breast cancer surgery with frozen section exam (Mayo Clinic Rochester). We added it in Introduction. 

2. A total of 19 studies (42 references) were analyzed; not sure if number of the studies is enough for this metanalysis.

We perform a rigorous search strategy to find the studies related to the frozen section in breast cancer surgery. We are confident that this search strategy covers all studies that could have available data to perform it. Here we have 19 studies including 6760 participants.

3. English needs to be polished; there are some grammatical errors.

Thank you. We performed a grammar review with a specialist. 

# REVIEWER 2: 

1. Some hospitals routinely perform intraoperative margin assessment in breast conserving surgery, while others do not. Can you compare advantage and disadvantage of different techniques, including gross analysis, radiology, and cytology examination? How is local recurrence and reoperation rate individually? What scenarios do authors prefer to do frozen sections rather than other methods?

 We added the paragraph in Introduction:

Despite the advantages of macroscopic analysis, this procedure can be performed directly by the surgeon, and boasts of higher accuracy (80%), sensitivity (49%) and specificity (86%) than other techniques.

2. Is there any difference about false positive rate and false negative rate among different cancer type, such as IDC, ILC or mucinous carcinoma?

We added a paragraph at 3.1.1 Sensitivity and subgroup analysis:

The sensibility analysis by histological subtype was not possible due to lack of individual data on each test. Only two authors performed an evaluation by histological type, which will be describe in the results. Osako et al. showed an increase of 11.9 chance of positive margins in the final pathology (p=0.01) in patients with invasive lobular carcinoma, larger tumors, or extensive intraductal component (EIC), and who were 50 years old or younger. Jorn et al. claimed that only disease multifocality (histologically discrete tumors at least 2 cm apart) could be a risk factor to increased reoperation rates, with OR of 3.41 (CI 1.38-8.40, p=0.008). The article did not associate histological subtype and tumor sizer with further surgeries. The invasive ductal carcinoma subtype had an OR of 0.75 (CI 0.31-1.82, p=0.37), invasive lobular carcinoma subtype had an OR of 2.29 (CI 0.52-9.98, p=0.37) and larger tumor size (> 2 cm) OR 1.33 (CI 0.26-6.74, p=0.733).

3. There is the large variability in false negative rate from 0% to 23%. What are reasons to cause this?

The reason for the large variability could be associated with the pathologist's experience. It is not possible to identify in the studies the physician's expertise. 

Moreover, a different definition of free margin between studies could explain this large variability.

4. In Jorns and Ko studies, the number of patients with reoperation is larger than the number of patients with false negative rates. Except false negative margin, is there any other reason to let surgeons re-operate patients?

Jorns study: 11 patients were reoperated. Among those 5 were true positive and 6 were false negative at frozen section which led the patient to another surgery. Based on this all patients were submitted to another surgery based only in the frozen section exam. 

 Ko study: considered some frozen sections as undetermined margins. This kind of margins were not evaluated even as positive neither negative. This factor increases the number of reoperation rate.

5. Is it possible to list turnaround time for results in the table considering that one of disadvantages of intraoperative assessment is to increase surgery time?

Eleven out of 19 studies described turnaround time to perform the frozen section exam, it ranges from 10 to 50 minutes.

We added a column in table 1.

Author Turnaround time

Anila 2016 20 min

Caruso 2011 20 min

Cendán 2005 13 min

Dener 2009 25 min

Ikeda 1997 -

Jorns 2014 24 min

Kikuyama 2015 -

Kim 2016 -

Ko 2017 40 min

Noguchi 1995 -

Nowikiewicz 2019 15 min

Olson 2007 25 min

Osako 2015 50 min

Pinotti 2002 -

Riedl 2008 20-30 min

Rusby 2008 10 -20 min

Sauter 1994 -

Tan 2014 -

Weber 2008 -

---

## [Editor Report · Decision Letter 2]

5 Mar 2021

Accuracy of frozen section in intraoperative margin assessment for breast-conserving surgery: a systematic review and meta-analysis

PONE-D-20-35344R2

Dear Dr. garcia,

We’re pleased to inform you that your manuscript has been judged scientifically suitable for publication and will be formally accepted for publication once it meets all outstanding technical requirements.

Kind regards,

Lanjing Zhang, MD, MS

Academic Editor

PLOS ONE

Additional Editor Comments (optional):

Sorry for missing your original rebuttal to the editor, but thank you for additional revision.

Please change "margins" in the first sentence of the abstract to "margin" during the copy editing.

Congratulations!

Reviewers' comments:

NA

---

## [Editor Report · Acceptance letter]

11 Mar 2021

PONE-D-20-35344R2 

Accuracy of frozen section in intraoperative margin assessment for breast-conserving surgery: a systematic review and meta-analysis 

Dear Dr. Garcia:

I'm pleased to inform you that your manuscript has been deemed suitable for publication in PLOS ONE. Congratulations! Your manuscript is now with our production department. 

Kind regards, 

on behalf of

Dr Lanjing Zhang 

Academic Editor

PLOS ONE